# Knowledge, Attitudes, and Practices of Influenza Vaccination among Parents of Children with Asthma: A Cross-Sectional Study

**DOI:** 10.3390/vaccines11061074

**Published:** 2023-06-07

**Authors:** Walid Al-Qerem, Anan Jarab, Alaa Hammad, Fawaz Alasmari, Jonathan Ling, Enas Al-Zayadneh, Montaha Al-Iede, Badi’ah Alazab, Leen Hajeer

**Affiliations:** 1Department of Pharmacy, Faculty of Pharmacy, Al-Zaytoonah University of Jordan, Amman 11733, Jordan; alaa.hammad@zuj.edu.jo (A.H.); b.alazab@zuj.edu.jo (B.A.); 2Department of Clinical Pharmacy, Faculty of Pharmacy, Jordan University of Science and Technology, Irbid 22110, Jordan; asjarab@just.edu.jo; 3College of Pharmacy, Al Ain University, Abu Dhabi 64141, United Arab Emirates; 4Department of Pharmacology and Toxicology, College of Pharmacy, King Saud University, Riyadh 12372, Saudi Arabia; ffalasmari@ksu.edu.sa; 5Faculty of Science and Wellbeing, University of Sunderland, Sunderland SR1 3SD, UK; jonathan.ling@sunderland.ac.uk; 6Department of Pediatrics, School of Medicine, University of Jordan, Amman 11910, Jordan; e.alzayadneh@ju.edu.jo (E.A.-Z.); m.al-iede@ju.edu.jo (M.A.-I.); 7School of Medicine, University of Jordan, Amman 11910, Jordan; lyn0182743@ju.edu.jo

**Keywords:** asthma, exacerbation, influenza, influenza vaccine, morbidity

## Abstract

Asthma is the most common chronic disease in childhood. Exacerbation is a significant problem for asthmatic patients, and viral infections remain the most frequent triggers of asthma exacerbations. This study explored knowledge, attitudes, and practices (KAP) of parents of asthmatic children towards providing influenza vaccine to their children. This cross-sectional study enrolled parents of asthmatic children who visited the outpatient respiratory clinics of two Jordanian hospitals. The present study enrolled 667 parents of asthmatic children (62.8% female). The median age of the participants’ children was 7 years. The results showed that 60.4% of the children with asthma never received a flu vaccine. Most of those who had received the flu vaccine reported that the side effects were mild (62.7%). Asthma duration was positively and significantly associated with increased vaccine hesitancy/rejection (OR = 1.093, 95% CI = (1.004–1.190), *p* = 0.04; and OR = 1.092, 95% CI = (1.002–1.189), *p* = 0.044, respectively). As the attitude towards flu vaccine score increases, odds of vaccination hesitancy/rejection decreased (OR = 0.735, 95% CI = (0.676–0.800), *p* < 0.001; and OR = 0.571, 95% CI = (0.514–0.634), *p* < 0.001, respectively). The main reasons for vaccination hesitancy/refusal included “I don’t think my child needs it” (22.3%) followed by “I forget it” (19.5%). The rate of vaccination among children was low and emphasized the necessity of encouraging parents with asthmatic children to vaccinate their children by conducting health awareness campaigns and also emphasized the role of doctors and other healthcare professionals.

## 1. Introduction

Asthma is defined by Global Initiative for Asthma (GINA) as chronic airway inflammation that is accompanied by cough, wheezing, chest tightness, and breathlessness [1]. Asthma is the most common chronic respiratory disease in children [2]. Prevalence of asthma in Jordanian children was estimated to be 10% in Amman and 16.6% in Ma’an governorate, which is comparable to the numbers reported worldwide [3,4].

Exacerbation is a major component of asthma, which is defined by the most recent (2022) GINA guidelines as an acute or sub-acute worsening in symptoms and lung function from the patient’s usual status, or in some cases, a patient may present for the first time with an exacerbation [1]. Uncontrolled asthma is associated with direct medical costs that were estimated to be 396 JD per child annually, and indirect medical costs on children and their families, such as school absence (2.48 days per asthmatic child), costs the caretaker $285 per asthmatic child [5,6].

Evidence of association between upper respiratory tract viral infections and asthma exacerbations in children is well established. Viral infections account for the vast majority of asthma attacks in children (>80%) [7]. CDC recommends that all people aged 6 months and older be vaccinated against flu yearly starting from September till the end of October, i.e., before the usual period of seasonal flu. People with asthma are at higher risk of developing complications from flu, such as worsening of symptoms, increasing mortality, and risk of developing pneumonia [8]. Although vaccination against influenza may have serious complications, such as myocarditis and pericarditis, the benefits of receiving the vaccine overweighs the risks, particularly among high-risk populations including those with asthma [9].

Although several studies have evaluated parental knowledge, attitude, and practices (KAP) towards influenza vaccine among parents of children with asthma, such studies in Jordan and the adjacent region are lacking. This study is the first to explore parental knowledge, practices, and intention towards flu vaccination in children with asthma in Jordan. Findings from this study will examine the importance of flu vaccine as a protective measure against developing asthma complications, with the aim of improving morbidity and mortality of children with asthma and prevent exacerbations caused by flu infection.

## 2. Materials and Methods

### 2.1. Methodology

The current cross-sectional study was conducted in the outpatient respiratory clinics at Jordan University Hospital in Amman and King Abdullah Hospital in Irbid in the period between October 2021 and January 2023. Inclusion criteria included children aged 18 or younger with diagnosis of asthma based on the Global Initiative for Asthma (GINA) guidelines [10]. The research pharmacist approached the parents of children who matched the inclusion criteria after being identified by the COPD nurse specialist. The research pharmacist provided the parents with an information sheet describing the study details and purposes. The researcher emphasized that participation is voluntary, and the parents had the right to refuse or withdraw from the study at any time without any effect on healthcare of the children. Parents who agreed to participate were asked to sign a consent form. The interview was conducted in a separate room at the outpatient clinic and took an average of 10 min to complete. The study was conducted according to the ethical principles of the Declaration of Helsinki. Ethical approval was obtained from Al-Zaytoonah University (Ref#22/23/2020–2021), The University of Jordan (Ref#2021–61), and Jordan University of Science and Technology (Ref#2021/07).

### 2.2. Data Collection and Study Instruments

Demographic information was obtained from the participants using a custom-designed questionnaire developed after an extensive literature review [11,12]. The present survey questionnaire is based on previously published studies by Al-Qerem et al. that evaluated parental acceptance of COVID-19 vaccine [13] after performing the necessary modifications for it to be suitable for the influenza vaccine among asthmatic children. 

Content validity was evaluated by an expert panel that included two pediatric pulmonologists, an infectious diseases specialist, and a clinical pharmacist. The questionnaire was translated/back translated from English to Arabic by different linguistic experts and the two versions were comparable. A pilot study that included 31 participants confirmed legibility of the questionnaire and their data were excluded from the statistical analysis. 

The questionnaire included seven domains assessing sociodemographics, medical information about children with asthma, knowledge about asthma, knowledge about flu and flu vaccines, and attitudes and practices towards vaccinating children against flu. The demographic data included two parts. The first part was for the child, which included age, gender, asthma duration, and exposure to passive smoking. The second part was for parental demographics, which included age, gender, and socioeconomic status. The second part of the questionnaire focused on the children’s medical information. Collected information included medications used, hospitalization during the last year, and asthma control assessed using the validated Arabic version of Asthma Symptoms Control (ASC) [14].

The third part evaluated knowledge about asthma and included whether asthma is contagious or hereditary, exacerbations, spirometry tests, different environmental triggers, and whether the disease gets worse when other diseases are present, among others. The fourth part evaluated knowledge about flu and included whether flu is same as the common cold and the cause and treatment of flu. The fifth part evaluated knowledge about the flu vaccine and included whether there is a vaccine against flu, the appropriate time to take the vaccine, types of flu vaccine, the number of doses, and the side effects of the flu vaccine. The sixth part assessed the attitude of parents towards vaccinating their children against flu using a five-point Likert scale (strongly agree, agree, neutral, disagree, or strongly disagree). Statements included “My child must receive the flu vaccine” as well as other statements related to complications of the vaccine, child’s pediatrician’s opinion regarding the vaccine, the effectiveness of the flu vaccine, and the chances of developing flu after receiving the vaccine. Reverse scoring was used for negatively worded questions. The final part included practices towards vaccinating children against flu that will evaluate flu vaccination habits, intention for future vaccination, and barriers against the flu vaccine. The knowledge and practice scores were calculated based on the participants’ answers to the designated questions.

### 2.3. Sample Size Calculation

Convenience sampling technique was applied based on a 95% significance level and a 5% margin of error to measure the required sample size. The minimum required sample was 385 participants. The current study included 667 participants [15]. 

### 2.4. Statistical Analysis

The data were analyzed using SPSS version 26.0. All categorical variables were presented as frequencies and percentages and continuous variables as medians (95% Cl). Cronbach’s alphas were calculated to evaluate the internal consistency of the questionnaire. Multinomial logistic regression was applied to determine the significant factors associated with patient intentions regarding influenza vaccination, and variables with *p*-value < 0.05 were considered statistically significant.

## 3. Results

The present study enrolled 667 parents of asthmatic children (62.8% female). Their median age was 38 (38–39) years. The majority of the participants were married (94.8%) and in the low-income group (86.9%), and 69.6% of them were in the high-education group. Many of the children with asthma were male (64.8%) with a median age of 7 (7–8) years. Half of the children with asthma have previously been hospitalized due to asthma (55.8%) with an asthma duration of 3.6 years (3–4) (Table 1).

Participants’ responses to asthma knowledge items are presented in Table 2. The median asthma knowledge score was 14 (14–15) out of a maximum possible score of 19. The highest correct responses were for the items “Does dust/air pollution trigger asthma symptoms?” and “Does smoking trigger asthma symptoms?” (97.5% and 95.1%, respectively). The poorest scores were for items “Does aspirin worsen asthma symptoms?” and “Do you know how to properly use the peak flow meter?” (12.7% and 15%, respectively). Internal consistency of the asthma knowledge scale was evaluated using computing Cronbach’s alpha, and the results indicated acceptable internal consistency (Cronbach’s alpha = 0.80).

Participants’ responses for flu vaccine knowledge are shown in Table 3. The median flu vaccine knowledge score was 4 (4–5) out of a maximum possible score of 9. The highest correct response was for item “Flu can spread from one person to another” (97.8%), while the least correct items were “When is the appropriate age to take the flu vaccine?” and “Do you know the difference between triple and quadruple flu vaccines?” (14.5% and 14.8%, respectively) (Cronbach’s alpha = 0.62).

Participants’ attitudes to the flu vaccine attitude items are presented in Table 4. The median flu vaccine attitude score was 23 (23–24) out of a maximum possible score of 35. In the reverse-coded statements, the highest disagree/strongly disagree was “I believe that my child gets sick because of the flu shot” (43.7%), while the least was “The flu vaccination may cause complications/troubles for my child” (29.5%). For the other statements, the highest agree/strongly agree responses were for “It is easy to reach the pharmacy /hospital to receive the flu vaccination” (82.8%), while the least was for “My child’s pediatrician believes that my child should receive the flu vaccine” (45.3%) (Cronbach’s alpha = 0.71).

Participants’ practices for asthma were evaluated using 10 statements. The most always/usually provided practice was “How often do you change your child beddings weekly?” (84.4%) followed by “How often does your child avoid exposure to dust/air pollution?” and “How often does your child avoid exposure to smoking?” (73.7% and 73.5%, respectively). The least always/usually provided practice were using flow meter at home and using air dehumidifier (7%, and 12.9% respectively). The median asthma practices score was 34 (34–35) out of a maximum possible score of 50 (Cronbach’s alpha = 0.77; Table 5).

Participants’ responses to flu vaccine items are presented in Table 6. The results revealed that 60.4% of the children with asthma never received a flu vaccine. Most of those who had received the flu vaccine reported that the side effects were mild (62.7%). The most reported side effect was fatigue (18.6%) followed by redness and fever (7.9% and 7.8%, respectively), while the least reported side effect was nausea (1.9%).

Multiple regression was performed to assess the association between sociodemographic variables and the intention to vaccinate against flu this year. Analysis revealed that as the asthma duration increases, hesitancy and the rejection to receive flu vaccine increases (OR = 1.093, 95% CI = (1.004–1.190), *p* = 0.04; and OR = 1.092, 95% CI = (1.002–1.189), *p* = 0.044, respectively). As the attitude toward flu vaccine score increases, the odds of vaccination hesitancy or rejection decrease (OR = 0.735, 95% CI = (0.676–0.800), *p* < 0.001; and OR = 0.571, 95% CI = (0.514–0.634), *p* < 0.001, respectively). Being in the low-education group decreased the odds of being hesitant toward receiving the flu vaccine compared to the high-education group (OR = 0.443, 95% CI = (0.256–0.765), *p* = 0.004; and OR = 0.520, 95% CI = (0.276–0.978), *p* = 0.042, respectively). 

Parents of children who had never received, received once, or received the vaccine once but not annually had higher odds of being hesitant or rejecting vaccination compared to parents of children who received the vaccine annually (hesitancy: OR = 24.336, 95% CI = (6.733–87.963), *p* <0.001; OR = 7.947, 95% CI = (2.080–30.369), *p* = 0.002; and OR = 8.454, 95% CI = (2.132–33.521), *p* = 0.002, respectively and rejection: OR = 36.425, 95% CI = (6.782–195.628), *p* < 0.001; OR = 10.947, 95% CI = (1.851–64.745), *p* = 0.008; and OR = 10.182, 95% CI = (1.694–61.196), *p* = 0.011, respectively). Moreover, parents of children who had uncontrolled or partially controlled asthma showed less odds of rejecting flu vaccination compared to parents of those who had controlled asthma (OR = 0.310, 95% CI = (0.146–0.655), *p* = 0.002; and OR = 0.393, 95% CI = (0.205–0.755), *p* = 0.005, respectively). Finally, as the frequency of hospitalization increases, the likelihood of being rejective increases (OR = 1.835, 95% CI = (1.032–3.263), *p* = 0.039) (Table 7).

Reasons for not wanting to vaccinate children with asthma are presented in Figure 1. The highest reported reason for non-vaccination was “I don’t think my child needs it” (22.3%) followed by “I forget it” (19.5%), while the least reported reason was “I don’t know the benefit of it” (4%).

## 4. Discussion

In this study, and for the first time, we evaluated parental perceptions towards the use of flu vaccination for children with asthma in Jordan. We found that 55.8% of children had been hospitalized due to uncontrolled asthma. This is similar to earlier studies that found that uncontrolled asthma prevalence ranged from 37% to 64% in primary care settings for children with asthma [16,17]. However, in a study on 2429 children between the ages of 4 and 17 at 29 pediatric care facilities across the United States [18], 46% of children had uncontrolled asthma, which was less than our finding stated. 

In this study 73.6% of parents exhibited “good knowledge” in asthma (14 out of 17 total). According to ad hoc criteria used in earlier studies, such as answering 50–60% of the questions correctly [19,20] or receiving a total score over the 50th percentile [21], respondents with high knowledge were identified. In previous studies, the proportion of “knowledgeable” caregivers (71.6% mothers and 67% fathers) was comparable to the percentage of caregivers seen in this study (73.6%), although lower (40–50%) than other studies [19,21]. Nevertheless, there were gaps in knowledge regarding the use of peak flow meters and the effect of aspirin on asthma symptoms. This is similar to previous studies that reported poor knowledge of peak flow meter usage and high rates of aspirin induced asthma [22,23].

The response patterns about asthma knowledge observed in this study were consistent with those observed in previous studies. Similar percentages were also discovered in other research with the knowledge of symptom causes such flu infections [24], allergens [25], air pollution [26], and cigarette smoke [25] (about 70%, 80%, 60%, and 70%, respectively). We found 90% of parents knew that cold weather triggered asthma. This finding is higher than previous studies’ findings, which reported that only about 40% of mothers thought that jogging in cold weather could cause an asthma attack [27,28]. According to earlier findings [21,29], the majority of the mothers (60–80%) had general knowledge about asthma and said they tried to protect their children from exposure to triggering factors. Finally, few pertinent questions about the management of spirometer test were included in questionnaires that had already been developed [30].

This current study showed that parents of children with asthma displayed good asthma practice management in avoiding asthmatic triggers, such as changing bedding, exposure to pets, exposure to cold weather, perfume, pollutants, smoking and different medications in parallel with good asthmatic knowledge. This corresponds to earlier research suggesting that parental knowledge may influence asthma management practices [19]. Previous work showed that almost all (97%) of the mothers were aware that exposure to cigarette smoke could exacerbate their child’s asthma, a small-er proportion (79%) said they always tried to protect their children from it [28]. This finding would be consistent with a prior survey in which 98% of parents agreed that smoking in close proximity to their children with asthma was harmful to their health, but lower percentages of parents stated reported that they never permit smoking at home (88%) or in the car (82%) [31].

Despite amazing achievements like the eradication of smallpox, some individuals continue to question the advantages of vaccines and worry about their safety. Ironically, the industrialized world and its upper socioeconomic levels are where vaccination skepticism and anti-vaccine beliefs are most prevalent [32]. All age groups experience high rates of influenza virus infection, but young children experience the highest rates of influenza virus isolation and infection load [33]. In the United States, influenza, and pneumonia cause 20,000 fatalities annually [34]. Children with chronic respiratory conditions, such as asthma, are more likely to experience problems from the flu. According to previous studies on children with asthma, viral infections can be the reason for up to 63–80% of asthma or wheezing flare-ups [35,36]. Hence, the recommendation of influenza vaccine for asthmatic children. In this study, we found that only 10.5% of children are vaccinated annually and 62.7% reported only mild side effects. These results parallel those of previous work conducted in Turkey where only 12% of asthmatic children are vaccinated annually [12]. In a study conducted in the USA on 317,596 asthmatic children in 2015–18, fewer than half of children with asthma received influenza vaccinations and mainly children with insurance [37]

In recent years, there has been increasing awareness of the importance of influenza vaccination for individuals with asthma. Children with asthma are at a higher risk of developing severe complications from the flu, including hospitalization and even death. However, despite the recommended annual flu vaccination for children with asthma [38], coverage remains suboptimal. This can be due to several reasons, including parental perceptions towards the flu vaccine [39]. As for flu vaccine knowledge, attitudes and practices, we found poor knowledge and practices but good attitudes toward flu vaccine (44.4%, 11% and 65.7% respectively). These results parallel those of other studies, which indicate that adult acceptance of vaccines is highly correlated with perceived effectiveness, personal experiences with vaccines, negative side effects, and prior vaccination [40]. Previous work conducted a group discussion expressed worries about the efficacy and safety of vaccines [41]. Several studies evaluating parental attitudes about childhood immunization reported that parents have voiced comparable worries about efficacy and safety of the vaccines [42]. Prior to an H1N1 vaccine campaign, a survey of healthcare professionals in Kenya revealed vac-cine side effects were a barrier to immunization [43]. Moreover, studies have reported the causes and remedies for influenza virus infection in a wide range of responses from respondents. Participants mentioned low ambient temperatures, dust, dirt, and smoke as causes of influenza during the pre-vaccination conversations. The first is linked to influenza transmission worldwide [44,45], whereas the latter three have not. The men-tion of dust and dirt, however, may be a sign that people are aware that air pollution from cooking and poor hygiene can accelerate the spread of disease. In this study, poor knowledge was associated with the cause of influenza in which most parents thought that bacteria were the main cause.

In a multinomial regression model, several factors were found to be associated with the intention to vaccinate against flu. Asthma duration and education of the parents were strongly correlated with children’s vaccination uptake. These results are comparable to those of cross-sectional studies carried out in Taiwan and Saudi Arabia, where half the participating parents were female and had completed at least middle school [46,47]. No positive association was found between parental knowledge of the value of children immunization with the decision to vaccinate. This parallels previous work [47]. However, a web-based study by Flood et al. revealed that parental knowledge of the value of children immunization was connected to the decision to vaccinate [48]. Knowing about the flu increased parental propensity to vaccinate their children. This might be due to the presence of additional considerations, such as availability of healthcare providers, that influence the decision to get vaccinated. An-other factor is attitude towards flu vaccination, with higher attitude scores associated with greater likelihood of a parent vaccinating their children. This is similar to a previous study by Chen et al. that used the health benefit model to find disparities in vaccination rates between Caucasian children and those of other ethnic groups. They found that parents of African Americans had greater reservations about the flu shot and were concerned that it would result in an illness [49]. Parents who did not vaccine annually were less likely to vaccinate their children. This may be due to the perceived benefits of the vaccine and the perceived risks of non-vaccination. Parental under-standing of the possible advantages of vaccination were positively correlated with the desire to vaccinate, according to a web-based study including Chinese parents [50]. A similar relationship between high vaccination uptake and the perceived advantages of vaccination was reported in another study conducted in five European nations [51]. This connection was reported for parents of healthy children and not those with asthma. Although aware of the advantages, parents of children with asthma may be reluctant to vaccinate them. Another study reported that even after controlling for other factors in the multivariate analysis, perceived benefits remained associated with vac-cine uptake among children with asthma. Moreover, parents who felt the flu infection is a risk for deteriorating asthma symptoms were more likely to have good attitudes toward the vaccine uptake [47].

The reasons for non-vaccination that were discovered among asthmatic kids in Jordan were consistent with the findings from other studies. According to research by Daley et al., frequent reminders improved parents’ likelihood of immunizing their children [52]. Additionally, American parents stated that other factors, such as accessibility and availability of care, played a role in their decision to vaccinate their children [49]. Regardless of vaccination uptake, the majority of Jordanian parents disagreed that a vaccine might harm children. It is necessary to conduct further qualitative study to deter-mine the precise aspects that influence Jordanian parents’ choices. Additionally, research demonstrates that parental social support from friends and family plays a critical role in influencing parents’ vaccination decisions among some Ameri-can parents [49]. According to Allison et al., parents who were persuaded by their friends’ or relatives’ positive comments about the vaccine were more likely to vaccinate their children [53]. The social support that healthcare professionals offer is another crucial reason. Previous research linked social support from healthcare professionals to the vaccination rate [54,55].

Gaglani et al.’s study revealed that a doctor’s advice had a significant influence on parents’ choices regarding flu vaccination [56]. Importantly, campaigns should be initiated on the importance of influenza vaccines for asthmatic children. These types of campaigns should be targeted to parents, healthcare providers as well as the public and should focus on the highlighted knowledge and practice gaps including vaccina-tion against influenza, aspirin administration and use of peak flow meter.

### Study Strength and Limitations

This study evaluated KAP towards influenza vaccine acceptance of parents with asthmatic children. We evaluated the prevalence of vaccinations among asthmatic children and examined the reported side effects associated with the vaccines. We also found variables associated with vaccination practices and barriers against receiving the influenza vaccine. Finally, the study evaluated children’s asthma severity using ASC, which is the recommended tool to evaluate asthma control in GINA guidelines. This is a crucial factor in determining their vaccination status and overall management. Nevertheless, there are several limitations in the present study. First, the data was based on self-reported questionnaire, which can be subject to social desirability and recall biases. Nevertheless, the low vaccination rates reported in the present study can increase the confidence of our findings. However, future work may adopt objective measures to validate the self-reported data and to provide a more accurate picture of parental perceptions and practices. Furthermore, the study was only conducted in two hospitals. While these hospitals serve large numbers of patients and cover a large geographic area, the inclusion of more medical centers could increase the generalizability of the study findings. Lastly, the study did not include qualitative data that could have provided a deeper understanding of the parents’ beliefs and perceptions about asthma and flu vaccination.

## 5. Conclusions

The rate of vaccination among children was low despite most parents having good knowledge and attitudes toward asthma and good perceptions of the vaccine against the flu in children with asthma. This study is one of few rare studies assessing how parents feel about immunizing their children with asthma against the flu to be carried out in the Middle East. One limitation of our study is that we were only able to include patients from two tertiary healthcare facilities. Similar surveys should be conducted at additional institutions in future studies to gain a deeper knowledge of the relationship between parental perceptions and vaccine uptake in Jordan. The results of the present study emphasize the necessity of encouraging parents with asthmatic children to vaccinate their children annually against the flu by facilitating the vaccination process and conducting health awareness campaigns.

## Figures and Tables

**Figure 1 vaccines-11-01074-f001:**
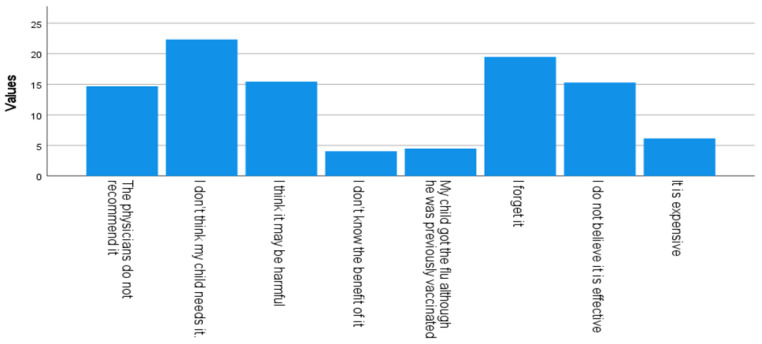
Reasons for non-vaccination.

**Table 1 vaccines-11-01074-t001:** Sociodemographic characteristics of children with asthma and their parents.

		Median (95% Lower-Upper CL) or Frequency (%)
Child’s age		7 (7–8)
Child’s sex	Female	235 (35.2%)
	Male	432 (64.8%)
Parent’s age		38 (38–39)
Parents’ sex	Female	419 (62.8%)
	Male	248 (37.2%)
Education	Low (High school or less)	203 (30.4%)
	High (Above high school)	464 (69.6%)
Marital status	Other	35 (5.2%)
	Married	632 (94.8%)
Income	<1000 JOD	573 (86.9%)
	>=1000 JOD	86 (13.1%)
Passive smoking	No	388 (58.2%)
	Yes	279 (41.8%)
Asthma duration		3.6 (3–4)
Previous asthma hospitalization	No	295 (44.2%)
	Yes	372 (55.8%)
Previous flu infection	No	82 (12.3%)
	Yes	585 (87.7%)

**Table 2 vaccines-11-01074-t002:** Participants’ responses to asthma knowledge items.

	No Frequency (%)	I Don’t Know Frequency (%)	Yes Frequency (%)
Asthma is an infectious disease *	608 (91.2%)	47 (7%)	12 (1.8%)
Asthma is a hereditary disease	135 (20.2%)	97 (14.5%)	435 (65.2%)
Patients may experience flare ups/exacerbations	14 (2.1%)	51 (7.6%)	602 (90.3%)
Asthma is a chronic disease	85 (12.7%)	46 (6.9%)	536 (80.4%)
Do you know the spirometry test?	188 (28.2%)	118 (17.7%)	361 (54.1%)
Do you know how to properly use the peak flow meter?	375 (56.2%)	192 (28.8%)	100 (15%)
Does cold weather trigger asthma symptoms?	34 (5.1%)	33 (4.9%)	600 (90%)
Does smoking trigger asthma symptoms?	14 (2.1%)	19 (2.8%)	634 (95.1%)
Does exposure to pets trigger asthma symptoms?	69 (10.3%)	79 (11.8%)	519 (77.8%)
Does perfume trigger asthma symptoms?	37 (5.5%)	24 (3.6%)	606 (90.9%)
Do dust/air pollution trigger asthma symptoms?	8 (1.2%)	9 (1.3%)	650 (97.5%)
Does pollen trigger asthma symptoms?	59 (8.8%)	83 (12.4%)	525 (78.7%)
Does allergic rhinitis make asthma symptoms worse?	37 (5.5%)	48 (7.2%)	582 (87.3%)
Does sinusitis make asthma symptoms worse?	26 (3.9%)	86 (12.9%)	555 (83.2%)
Does flu make asthma symptoms worse?	29 (4.3%)	37 (5.5%)	601 (90.1%)
Does tonsilitis make asthma symptoms worse?	73 (10.9%)	100 (15%)	494 (74.1%)
Do you/your child know how to correctly use asthma inhalers?	81 (12.1%)	21 (3.1%)	565 (84.7%)
Does aspirin worsen asthma symptoms?	99 (14.8%)	483 (72.4%)	85 (12.7%)
Do NSAIDS (brofen, voltaren) worsen asthma symptoms?	80 (12%)	467 (70%)	120 (18%)

* The correct answer for this item is no.

**Table 3 vaccines-11-01074-t003:** Participants’ responses to flu vaccine knowledge items.

		Frequency (%)
Is there a vaccine against flu?	No	43 (6.4%)
	I don’t know	50 (7.5%)
	Yes *	574 (86.1%)
Do you know the difference between triple and quadruple flu vaccines?	No	303 (45.4%)
	I don’t know	265 (39.7%)
	Yes *	99 (14.8%)
Does the vaccine have side effects?	No	128 (19.2%)
	I don’t know	354 (53.1%)
	Yes *	185 (27.7%)
Flu is caused by bacteria.	No *	357 (53.5%)
	I don’t know	142 (21.3%)
	Yes	168 (25.2%)
Flu can spread from one person to another.	No	8 (1.2%)
	I don’t know	7 (1%)
	Yes *	652 (97.8%)
Antibiotics can be used to treat flu.	No *	250 (37.5%)
	I don’t know	50 (7.5%)
	Yes	367 (55%)
When is the appropriate time to take the flu vaccine?	January–March	81 (12.1%)
	November–December	83 (12.4%)
	September–October *	364 (54.6%)
	I don’t know	139 (20.8%)
When is the appropriate age to take the flu vaccine?	At birth	15 (2.2%)
	>=6 months *	97 (14.5%)
	At no specific age	460 (69%)
	I don’t know	95 (14.2%)
What is the maximum limit of flu vaccine shots a child can receive per visit?	One dose *	282 (42.3%)
	Two doses	18 (2.7%)
	Three doses	15 (2.2%)
	I don’t know	352 (52.8%)

* Correct answers for the items.

**Table 4 vaccines-11-01074-t004:** Participants’ responses to flu vaccine attitude items.

	Strongly Disagree Frequency (%)	Disagree Frequency (%)	Neutral Frequency (%)	Agree Frequency (%)	Strongly Agree Frequency (%)
I believe that my child must receive the flu vaccination	14 (2.1%)	123 (18.4%)	204 (30.6%)	161 (24.1%)	165 (24.7%)
It is easy to reach the pharmacy /hospital to receive the flu vaccination	10 (1.5%)	35 (5.2%)	70 (10.5%)	268 (40.2%)	284 (42.6%)
My child’s pediatrician believes that my child should receive the flu vaccine	36 (5.4%)	150 (22.5%)	179 (26.8%)	120 (18%)	182 (27.3%)
Flu vaccination prevents infection by the flu virus	12 (1.8%)	107 (16%)	237 (35.5%)	201 (30.1%)	110 (16.5%)
The flu vaccination may cause complications/troubles for my child *	58 (8.7%)	139 (20.8%)	286 (42.9%)	124 (18.6%)	60 (9%)
I believe that my child gets sick because of the flu shot *	73 (10.9%)	219 (32.8%)	264 (39.6%)	79 (11.8%)	32 (4.8%)
I am worried about the chances of my child contracting the flu because of the flu vaccine *	76 (11.4%)	183 (27.4%)	246 (36.9%)	104 (15.6%)	58 (8.7%)

* Reverse-coded statements.

**Table 5 vaccines-11-01074-t005:** Participants’ responses to asthma practice items.

	Never	Rarely	Sometime	Usually	Always
	Frequency (%)	Frequency (%)	Frequency (%)	Frequency (%)	Frequency (%)
How often does your child avoid exposure to smoking?	13 (1.9%)	44 (6.6%)	120 (18%)	198 (29.7%)	292 (43.8%)
How often does your child avoid exposure to pets?	23 (3.4%)	68 (10.2%)	136 (20.4%)	170 (25.5%)	270 (40.5%)
How often does your child avoid exposure to cold weather?	12 (1.8%)	29 (4.3%)	151 (22.6%)	221 (33.1%)	254 (38.1%)
How often does your child avoid exposure to perfumes?	24 (3.6%)	46 (6.9%)	165 (24.7%)	223 (33.4%)	209 (31.3%)
How often does your child avoid exposure to dust/air pollution?	8 (1.2%)	40 (6%)	127 (19%)	241 (36.1%)	251 (37.6%)
How often do you change your child beddings weekly?	7 (1%)	20 (3%)	77 (11.5%)	171 (25.6%)	392 (58.8%)
How often do you avoid giving your child aspirin or NSAIDS?	72 (10.8%)	110 (16.5%)	198 (29.7%)	134 (20.1%)	153 (22.9%)
Does your child perform spirometry test on every visit to the physician?	189 (28.3%)	220 (33%)	117 (17.5%)	72 (10.8%)	69 (10.3%)
How often does your child use an air dehumidifier?	354 (53.1%)	145 (21.7%)	82 (12.3%)	44 (6.6%)	42 (6.3%)
How often does your child use a flow meter at home?	382 (57.3%)	167 (25%)	71 (10.6%)	27 (4%)	20 (3%)

**Table 6 vaccines-11-01074-t006:** Participants’ responses to flu vaccine practice items.

		Frequency (%)
How often does your child receive a flu vaccine?	Never	403 (60.4%)
	One time	116 (17.4%)
	More than one time but not annually	78 (11.7%)
	Annually	70 (10.5%)
What type of vaccine did your child receive?	Trivalent	61 (9.1%)
	Quadrivalent	40 (6%)
	I don’t know	566 (84.9%)
Reported side effects		
Fever	No	615 (92.2%)
	Yes	52 (7.8%)
Redness	No	614 (92.1%)
	Yes	53 (7.9%)
Fatigue	No	543 (81.4%)
	Yes	124 (18.6%)
Headache	No	643 (96.4%)
	Yes	24 (3.6%)
Nausea	No	654 (98.1%)
	Yes	13 (1.9%)
What was the severity of the side effects?	Mild	128 (62.7%)
	Moderate	64 (31.4%)
	Severe	12 (5.9%)

**Table 7 vaccines-11-01074-t007:** Multinomial regression model between different sociodemographic characteristics and the intention to vaccinate against flu this year.

				Maybe vs. Yes			No vs. Yes		
		*p*-Value	OR	95% Confidence Interval for OR			95% Confidence Interval for OR		
				Lower Bound	Upper Bound	*p*-Value	Lower Bound	Lower Bound	Upper Bound
Child’s age		0.891	0.996	0.936	1.059	0.807	1.008	0.944	1.077
Asthma duration		0.04	1.093	1.004	1.19	0.044	1.092	1.002	1.189
Asthma practice score		0.421	0.985	0.95	1.022	0.792	0.994	0.954	1.037
Asthma knowledge score		0.133	1.066	0.981	1.158	0.329	1.047	0.955	1.147
Flu vaccine attitude score		<0.001	0.735	0.676	0.8	<0.001	0.571	0.514	0.634
Parent’s age		0.825	1.004	0.969	1.04	0.7	0.992	0.954	1.032
Flu vaccine and flu knowledge score		0.387	0.937	0.809	1.086	0.204	1.111	0.944	1.307
Child sex	Female	0.306	0.765	0.459	1.276	0.124	1.561	0.886	2.752
	Male (REF)	.		.		.	.	.	.
Income	<1000 JOD	0.393	1.367	0.668	2.798	0.652	1.197	0.547	2.620
	>= 1000 JOD (REF)	.		.		.	.	.	.
Education	Low	0.004	0.443	0.256	0.765	0.042	0.52	0.276	0.978
	High (REF)	.		.		.	.	.	.
Control status	Uncontrolled	0.743	0.897	0.468	1.719	0.002	0.31	0.146	0.655
	Partial controlled	0.31	0.734	0.404	1.333	0.005	0.393	0.205	0.755
	Controlled (REF)	.		.		.	.	.	.
Parent sex	Female	0.344	1.292	0.76	2.195	0.549	0.835	0.463	1.506
	Male (REF)	.		.		.	.	.	.
Frequency of vaccination against flu	Never	<0.001	24.336	6.733	87.963	<0.001	36.425	6.782	195.628
	One time	0.002	7.947	2.08	30.369	0.008	10.947	1.851	64.745
	More than one time but not annually	0.002	8.454	2.132	33.521	0.011	10.182	1.694	61.196
	Annually (REF)	.		.		.	.	.	.
Previous hospitalization for asthma		0.809	1.066	0.633	1.796	0.039	1.835	1.032	3.263

## Data Availability

Available upon request.

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
