# Peer review of "Knowledge, Attitudes, and Practices of Influenza Vaccination among Parents of Children with Asthma: A Cross-Sectional Study"

_vaccines, 2023, doi:10.3390/vaccines11061074_

Round 1
Reviewer 1 Report
Thank you for the possibility to review this important article on hot topic:
Knowledge, Attitude, and Practices of Influenza Vaccination among Parents of Asthmatic Children: A Cross-sectional Study
I have some proposals:
1. In discussions please overview and compare your data with data from another countries from your region and also with another countries. I think it will be interesting for readers
2. Please present limitations of your study
3. To review discussions and to add more sources
Author Response
The authors would like to thank the the reviewer for her/his time and thorough evaluation of the manuscript. We believe that we have addressed all the raised comments which have substantially improved the quality of the manuscript.
Please see our detailed responses below
Reviewer(s)' Comments to Author:
Reviewer 1
- In discussions please overview and compare your data with data from another countries from your region and also with another countries. I think it will be interesting for readers
- The following has been added to the discussion section: “Despite amazing achievements like the eradication of smallpox, some individuals continue to question the advantages of vaccines and worry about their safety. Ironically, the industrialized world and its upper socioeconomic levels are where vaccination skepticism and anti-vaccine beliefs are most prevalent (31).
All age groups experience high rates of influenza virus infection, but young children experience the highest rates of influenza virus isolation and infection load (32). In the United States, influenza and pneumonia cause 20,000 fatalities annually (33). Children with chronic respiratory conditions, such as asthma, are more likely to experience problems from the flu. According to previous studies on children with asthma, viral infections can be to the reason for up to 63–80% of asthma or wheezing flare-ups (34,35). Hence, the recommendation of influenza vaccine for asthmatic children. In this study, we found that only 10.5% of children are vaccinated annually and 62.7% reported only mild side effects. These results parallel those of previous work conducted in Turkey where only 12% of asthmatic children are vaccinated annually (36). In a USA study, on 317,596 asthmatic children in 2015–18, fewer than half of children with asthma received influenza vaccinations and mainly children those without insurance (37)”
- Please present limitations of your study
-A section on the study strengths and limitations has been added to the discussion:
“This study evaluated KAP towards influenza vaccine acceptance of parents with asthmatic children. We evaluated the prevalence of vaccinations among asthmatic children and examined the reported side effects associated with the vaccines. We also found variables associated with vaccination practices and barriers against receiving the influenza vaccine. Finally, the study evaluated children's asthma severity, by applying the ASC which is the recommended tool to evaluate asthma control in GINA guidelines. This is a crucial factor in determining their vaccination status and overall management. Nevertheless, there are several limitations in the present study. First, the data was based on self-reported questionnaire, which can be subject to social desirability and recall biases. Nevertheless, the low vaccination rates reported in the present study can increase the confidence of our findings. However, future work may adopt objective measures to validate the self-reported data and to provide a more accurate picture of parental perceptions and practices. Furthermore, the study was only conducted in two hospitals. While these hospitals serve large numbers of patients and cover a large geographic area, including more medical centers could increase the generalizability of the study findings. Lastly, the study did not include qualitative data that could have provided a deeper understanding of the parents' beliefs and perceptions about asthma and flu vaccination.
- To review discussions and to add more sources
-The discussion has been modified, and new references added.
Reviewer 2 Report
The study provides valuable insights into the knowledge, attitudes, and practices of parents of asthmatic children towards providing influenza vaccine to their children.
However following comments should be addressed before the decision.
1- The abstract does not mention how the study's results could be used to develop effective interventions to increase vaccination rates among this population. It would be helpful to provide more information on how the findings could be used to design targeted interventions to address the reasons for vaccination hesitancy and improve vaccination rates among children with asthma.
2- Abstract is divided into different sections with headings, please read the guidelines of the journal carefully and correct it accordingly.
3- The introduction lacks a clear statement of the research gap or the specific objective of the study. It would be beneficial to clearly state the research question or objective of the study in the introduction to provide a focused direction for the readers and emphasize the novelty of the research.
4- The study's methodology seems well-designed and comprehensive. However, the study's sampling technique needs clarification. The authors have mentioned that they have used convenience sampling with a 95% significance level and a 5% margin of error to calculate the sample size, but they have not explained how they selected the participants. The authors should provide more information about how they approached and recruited the participants for this study. This information is crucial to establish the study's representativeness and generalizability.
5- The results indicate that the median asthma knowledge score was acceptable, but there were gaps in knowledge regarding the use of peak flow meters and the effect of aspirin on asthma symptoms. Furthermore, the study did not include the children's severity of asthma, which is a crucial factor in determining their vaccination status and overall management. Additionally, the study relied on self-reported practices, which may be subject to recall bias or social desirability bias. Furthermore, the study was conducted in a single center and a limited geographic area, which may limit the generalizability of the findings. Lastly, the study did not include qualitative data that could have provided a deeper understanding of the parents' beliefs and perceptions about asthma and flu vaccination.
6- In the discussion section, it would be helpful if the authors could discuss and justify why they considered this sample size adequate and discuss the implications of their findings for the general population. As the sample size of 178 participants might not be enough to generalize the findings to the entire population of parents of children with asthma in Jordan.
Secondly, the study relied on self-reported data, which can be subject to social desirability bias. The authors need to acknowledge this limitation and discuss the extent to which it may have influenced the results. Additionally, they may consider using objective measures to validate the self-reported data and to provide a more accurate picture of parental perceptions and practices towards flu vaccination for children with asthma in Jordan.
There are a few minor errors in sentences that could be corrected to improve the clarity of the writing. For example, in discussion, "stated finding" should be changed to "findings stated." Additionally, in another sentence, "stated that smoking is never permitted" could be changed to "reported that they never permit smoking." These changes would improve the sentence structure and make the meaning clearer. Therefore, authors should read the manuscript carefully and improve the clarity.
There are a few minor errors in sentences that could be corrected to improve the clarity of the writing. For example, in discussion, "stated finding" should be changed to "findings stated." Additionally, in another sentence, "stated that smoking is never permitted" could be changed to "reported that they never permit smoking." These changes would improve the sentence structure and make the meaning clearer. Therefore, authors should read the manuscript carefully and improve the clarity.
Author Response
The authors would like to thank the the reviewer for her/his time and thorough evaluation of the manuscript. We believe that we have addressed all the raised comments which have substantially improved the quality of the manuscript.
Please see our detailed responses below
Reviewer(s)' Comments to Author:
Reviewer2
The study provides valuable insights into the knowledge, attitudes, and practices of parents of asthmatic children towards providing influenza vaccine to their children.
-Thank you for this comment.
However following comments should be addressed before the decision.
- The abstract does not mention how the study's results could be used to develop effective interventions to increase vaccination rates among this population. It would be helpful to provide more information on how the findings could be used to design targeted interventions to address the reasons for vaccination hesitancy and improve vaccination rates among children with asthma.
-The following was added “The rate of vaccination among children was low, which emphasizes the necessity of encouraging parents with asthmatic children to vaccinate their children by conducting health awareness campaigns that emphasize the role of doctors and healthcare professionals.”
- Abstract is divided into different sections with headings, please read the guidelines of the journal carefully and correct it accordingly.
-The abstract was modified accordingly
3- The introduction lacks a clear statement of the research gap or the specific objective of the study. It would be beneficial to clearly state the research question or objective of the study in the introduction to provide a focused direction for the readers and emphasize the novelty of the research.
-The following was added to the introduction: “Although several studies have evaluated parental knowledge, attitude, and practices (KAP) towards influenza vaccine among parents of children with asthma, such studies in Jordan and the broader region are lacking. This study is the first to explore parental knowledge, practices and intention towards flu vaccination in children with asthma in Jordan. This study will examine the importance of the flu vaccine as a protective measure against developing asthma complications, with the aim of improving morbidity and mortality of children with asthma and prevent exacerbations caused by flu infection.”
4- The study's methodology seems well-designed and comprehensive. However, the study's sampling technique needs clarification. The authors have mentioned that they have used convenience sampling with a 95% significance level and a 5% margin of error to calculate the sample size, but they have not explained how they selected the participants. The authors should provide more information about how they approached and recruited the participants for this study. This information is crucial to establish the study's representativeness and generalizability.
-The following was added to the method section: “The research pharmacist approached the parents of children who matched the inclusion criteria after being identified by the COPD Nurse Specialist. The research pharmacist provided the parents with an information sheet describing the study details and purposes. The researcher emphasized that participation is voluntary and the with the right to refuse or withdraw from the study at any time without any effect on children healthcare. Parents who agreed to participate were asked to sign a consent form. The interview was conducted in a separate room at the outpatient clinic and took an average of 10 minutes to complete.”
5- The results indicate that the median asthma knowledge score was acceptable, but there were gaps in knowledge regarding the use of peak flow meters and the effect of aspirin on asthma symptoms.
-The following was added to the discussion: “Nevertheless, there were gaps in knowledge regarding the use of peak flow meters and the effect of aspirin on asthma symptoms. This is similar to previous studies that reported poor knowledge of peak flow meter usage and high rates of aspirin induced asthma (23,24)” and “Importantly, campaigns should be initiated on the importance of influenza vaccines for asthmatic children. These types of campaigns should be targeted to parents, healthcare providers as well as the public and should focus on the highlighted knowledge and practice gaps including vaccination against influenza, aspirin administration and use of peak flow meter.”
Furthermore, the study did not include the children's severity of asthma, which is a crucial factor in determining their vaccination status and overall management.
-Please note that the study evaluated children's severity of asthma, by applying the ASC which is the recommended tool to evaluate asthma control in the GINA guidelines and the results of the ASC are represented in the results and their association with vaccination practice was evaluated.
Additionally, the study relied on self-reported practices, which may be subject to recall bias or social desirability bias. Furthermore, the study was conducted in a single center and a limited geographic area, which may limit the generalizability of the findings. Lastly, the study did not include qualitative data that could have provided a deeper understanding of the parents' beliefs and perceptions about asthma and flu vaccination.
--Please note that the study included two hospitals. A section about study strengths and limitations has been added to the discussion which includes the following “This study evaluated KAP towards influenza vaccine acceptance of parents with asthmatic children. We evaluated the prevalence of vaccinations among asthmatic children and examined the reported side effects associated with the vaccines. We also found variables associated with vaccination practices and barriers against receiving the influenza vaccine. Finally, the study evaluated children's asthma severity, by applying the ASC which is the recommended tool to evaluate asthma control in GINA guidelines. This is a crucial factor in determining their vaccination status and overall management. Nevertheless, there are several limitations in the present study. First, the data was based on self-reported questionnaire, which can be subject to social desirability and recall biases. Nevertheless, the low vaccination rates reported in the present study can increase the confidence of our findings. However, future work may adopt objective measures to validate the self-reported data and to provide a more accurate picture of parental perceptions and practices. Furthermore, the study was only conducted in two hospitals. While these hospitals serve large numbers of patients and cover a large geographic area, including more medical centers could increase the generalizability of the study findings. Lastly, the study did not include qualitative data that could have provided a deeper understanding of the parents' beliefs and perceptions about asthma and flu vaccination.”
6- In the discussion section, it would be helpful if the authors could discuss and justify why they considered this sample size adequate and discuss the implications of their findings for the general population. As the sample size of 178 participants might not be enough to generalize the findings to the entire population of parents of children with asthma in Jordan.
-Please note that the study enrolled 667 parents of children with asthma which was substantially higher than the required sample size to conduct this study, as reported in the method section.
Secondly, the study relied on self-reported data, which can be subject to social desirability bias. The authors need to acknowledge this limitation and discuss the extent to which it may have influenced the results. Additionally, they may consider using objective measures to validate the self-reported data and to provide a more accurate picture of parental perceptions and practices towards flu vaccination for children with asthma in Jordan.
-As stated above, a limitation section that included the points raised by the reviewer, has been added to the manuscript
There are a few minor errors in sentences that could be corrected to improve the clarity of the writing. For example, in discussion, "stated finding" should be changed to "findings stated." Additionally, in another sentence, "stated that smoking is never permitted" could be changed to "reported that they never permit smoking." These changes would improve the sentence structure and make the meaning clearer. Therefore, authors should read the manuscript carefully and improve the clarity.
-Changes were made accordingly and the manuscript’s language was modified’s
Reviewer 3 Report
This is well designed and written cross sectional study by Walid Al-Qerem et al exploring knowledge, attitudes, and practices of parents of young asthmatic children toward vaccination against influenza. The survey included seven domains assessing socio-demographics, medical information about children with asthma, knowledge about asthma, knowledge about flu and flu vaccines and attitudes and practices towards vaccinating children against flu.
Enrollment included 667 parents of asthmatic children (62.8% female) of rather similar age of 38. Most of the participants were married (94.8%) and in the low-income group (86.9%), with 69.6% of them were in the high education group. Median age of children with asthma was 7 and half of the children has previously been hospitalized due to asthma. Importantly, the rate of vaccination among children was low, despite most parents having good knowledge and attitudes toward asthma and had good perceptions of the vaccine against the flu in children with asthma. This study is one of the rare studies looking into how parents feel about immunizing their children with asthma against the flu to be carried out in the Middle East. While author list limitations of the study and propose expanding it beyond two tertiary healthcare facilities one could expect to find proposed solutions to mitigate low vaccination rates based on presented study. Overall, this is well conducted and clearly presented study of hgh interest to health practitioners.
This certainly needs to be addressed by clear statement of the significance of the study (perhaps with proper outline of the hypothesis tested with what gap of knowledge it may answer). Given valuable data obtained through rather meticulously designed survey I do encourage authors for discussion of possible improvements and/or interventions to improve vaccination rates. Answering properly stated hypothesis would be very helpful here. Overall, this is well conducted and clearly presented study shedding important lights into demographics and reasoning of parents on decision to vaccinate their children against flu; however, some important conclusions are missing regarding possible remedies to the problem.
Author Response
The authors would like to thank the reviewer for her/his time and thorough evaluation of the manuscript. We believe that we have addressed all the raised comments which have substantially improved the quality of the manuscript.
Please see our detailed responses below
Reviewer(s)' Comments to Author:
Reviewer 3
This is well designed and written cross sectional study by Walid Al-Qerem et al exploring knowledge, attitudes, and practices of parents of young asthmatic children toward vaccination against influenza. The survey included seven domains assessing socio-demographics, medical information about children with asthma, knowledge about asthma, knowledge about flu and flu vaccines and attitudes and practices towards vaccinating children against flu.
-The authors would like to thank the reviewer for his endorsement
Enrollment included 667 parents of asthmatic children (62.8% female) of rather similar age of 38. Most of the participants were married (94.8%) and in the low-income group (86.9%), with 69.6% of them were in the high education group. Median age of children with asthma was 7 and half of the children has previously been hospitalized due to asthma. Importantly, the rate of vaccination among children was low, despite most parents having good knowledge and attitudes toward asthma and had good perceptions of the vaccine against the flu in children with asthma. This study is one of the rare studies looking into how parents feel about immunizing their children with asthma against the flu to be carried out in the Middle East. While author list limitations of the study and propose expanding it beyond two tertiary healthcare facilities one could expect to find proposed solutions to mitigate low vaccination rates based on presented study. Overall, this is well conducted and clearly presented study of hgh interest to health practitioners.
-Thank you for this the following was added to the limitation section: “Furthermore, the study was conducted in two hospitals only, although these hospitals serve large number of patients and cover large geographic area, including more medical centers can increase the generalizability of the study findings”
This certainly needs to be addressed by clear statement of the significance of the study (perhaps with proper outline of the hypothesis tested with what gap of knowledge it may answer). Given valuable data obtained through rather meticulously designed survey I do encourage authors for discussion of possible improvements and/or interventions to improve vaccination rates. Answering properly stated hypothesis would be very helpful here. Overall, this is well conducted and clearly presented study shedding important lights into demographics and reasoning of parents on decision to vaccinate their children against flu; however, some important conclusions are missing regarding possible remedies to the problem.
-The following was added to the introduction: “Although several studies have evaluated parental knowledge, attitude, and practices (KAP) towards influenza vaccine among parents of children with asthma, such studies in Jordan and the region are lacking. This study is the first to explore parental knowledge, practices and intention towards flu vaccination in children with asthma in Jordan. Findings from this study will examine the importance of flu vaccine as a protective measure against developing asthma complications, with the aim of improving morbidity and mortality of children with asthma and prevent exacerbations caused by flu infection” and the following was added to the discussion:” The reasons for non-vaccination that were discovered among asthmatic kids in Jordan were consistent with the findings from other studies. According to research by Daley et al., frequent reminders improved parents' likelihood of immunizing their children (54). Additionally, American parents stated that other factors, such as accessibility and availability of care, played a role in their choice to vaccinate their children (55). Regardless of vaccination uptake, the majority of Jordanian parents disagreed that a vaccine might harm children. It is necessary to conduct further qualitative study to deter-mine the precise aspects that influence Jordanian parents' choices. Additionally, research demonstrates that parental social support from friends and family plays a critical role in influencing parents' vaccination decisions among some American parents (55). According to Allison et al., parents who were persuaded by their friends' or relatives' positive comments about the vaccine were more likely to vaccinate their children (56). The social support that healthcare professionals offer is another crucial reason. Previous research, linked social support from healthcare professionals to the vaccination rate (57,58). Gaglani et al.'s found that a doctor's advice had a significant influence on parents' choices regarding flu vaccination (59). Importantly, campaigns should be initiated on the importance of influenza vaccines for asthmatic children. These types of campaigns should be targeted to parents, healthcare providers as well as the public and should focus on the highlighted knowledge and practice gaps including vaccination against influenza, aspirin administration and use of peak flow meter.”
Round 2
Reviewer 2 Report
The revised manuscript is now acceptable.